# Impact of Particle Size on the Setting Behavior of Tricalcium Silicate: A Comparative Study Using ISO 6876 Indentation Testing and Isothermal Induction Calorimetry

**DOI:** 10.3390/bioengineering11010036

**Published:** 2023-12-28

**Authors:** Anarela Vassen Bernardi, Marcelo Tramontin Souza, Oscar Rubem Klegues Montedo, Felipe Henrique Fassina Domingues, Sabrina Arcaro, Patrícia Maria Poli Kopper

**Affiliations:** 1Graduate Program in Dentistry, School of Dentistry, Federal University of Rio Grande do Sul (UFRGS), Porto Alegre 90010-150, RS, Brazil; anarela.bernardi@unesc.net (A.V.B.); patricia.kopper@ufrgs.br (P.M.P.K.); 2Grupo de Pesquisa de Biomateriais e Materiais Nanoestruturados, Laboratório de Cerâmica Técnica (CerTec), Universidade do Extremo Sul Catarinense (UNESC), Criciúma 88806-000, SC, Brazil; okm@unesc.net (O.R.K.M.); sarcaro@unesc.net (S.A.); 3Graduate Program in Science, Innovation and Modelling in Materials (PROCIMM), State University of Santa Cruz (UESC), Ilhéus 45662-900, BA, Brazil; 4Graduate Program in Dentistry, School of Dentistry, Federal University of Mato Grosso do Sul (UFMS), Campo Grande 79070-900, MS, Brazil; felipehfd@gmail.com

**Keywords:** tricalcium silicate, biocement, ISO 6876 indentation test, isothermal calorimetry

## Abstract

This study examines the impact of particle size on the setting behavior of tricalcium silicate powders. The setting behavior was evaluated using ISO 6876 indentation testing and isothermal induction calorimetry techniques. The objective was to compare the outcomes obtained from these methods and establish a correlation between particle size and setting characteristics. The cement pastes were manually mixed with a water-to-solid ratio of 0.66 for conducting indentation tests according to ISO 6876, while calorimetry measurements were performed using isothermal (conduction) calorimetry at room temperature. The findings demonstrate a significant influence of smaller particle sizes on accelerating the hydration process of cement pastes, resulting in a reduction of setting time by up to 24%. Moreover, the final setting times obtained through the indentation method closely approximate the inflection points of the acceleration curves acquired by calorimetry, with time deviations of less than 12% regardless of particle size.

## 1. Introduction

Bioceramic cements have gained widespread use in various dental applications, including root repair in teeth with furcal perforations, open apex, pulp necrosis, and surgical endodontic treatment [1,2]. They have also been suggested as filling materials [3]. As biocompatibility is a critical factor influenced by cement mineralogy, clinical performance, especially in biocement applications, hinges on particle size and temperature. These factors play a crucial role in determining the cement’s viscosity and setting time, directly impacting its ability to effectively penetrate and fill dentinal tubules and lateral canals [1,4,5,6]. Root canal sealers can exhibit varying viscosities, often assessed through flow measurements following the ISO 6876/2012 standard [7]. Certain endodontic sealers are specifically designed with higher fluidity to facilitate penetration into intricate canal systems, while others may possess lower fluidity to optimize handling characteristics. The ISO 6876/2012 recommends a minimum of 17 mm. In a study conducted by Torres et al. [8], different root canal sealers were found to exhibit values near 20 mm.

While the influence of temperature is difficult to determine due to variations in environmental and body temperatures, controlling particle size represents another means of manipulating the setting time to meet ideal clinical standards. The particle size of root canal sealers can vary based on the specific type used. Root sealer cements typically contain particles size under 10 μm [9,10], which aids in better flow and penetration into complex canal systems and contributes to achieving a reasonable setting time. From a practical standpoint, an excessively short setting time can negatively affect the workability of the cement, while an excessively long setting time can compromise cement solubility and disintegration in tissue fluid, thereby impairing sealing ability and bioactivity. In clinical practice, the setting time of endodontic cements is typically less than 30 min, normally achieved with chemical admixtures accelerating the setting time.

In dentistry, setting time plays a critical role, especially in luting materials where fluidity is essential, such as in root repairs and filling materials [11]. It is crucial to strike the right balance in setting time, avoiding excessively short times that impede proper flow, and excessively long times that may lead to solubilization and disintegration [12,13].

Setting time can be divided into two key phases: initial and final setting times. The initial setting time is the duration from the moment water is added to the cementitious mixture until the material starts to lose its plasticity and begins to harden. This stage is particularly important as it determines the window during which the material can be placed, molded, or worked on before it becomes too rigid. On the other hand, the final setting time represents the period from the introduction of water to the cementitious mixture until the material achieves its full hardening and load-bearing strength. Knowing the final setting time is essential for determining when the hardened material can be subjected to stresses safely. This occurs through the formation of a cross-linked network of hydration products, which can be evaluated and compared using indentation testing and calorimetric analysis.

The Gillmore needle penetration resistance test, based on ISO 6876, is commonly employed to measure the setting time of dental cements due to its simplicity [7]. This method aims to determine the setting time by measuring the penetration of a standardized weighted indenter through cement pastes. Although well-established, this technique lacks accuracy in revealing specific changes occurring during cement hydration. Additionally, it is limited by factors such as operator experience, maintenance of humidity and room temperature during indentation, and mold material characteristics. Consequently, indentation methods may yield different results for setting time, as exemplified by MTA (Angelus, Brazil), with a setting time of 145 min according to ISO 6876 [14] and 230 min according to ISO 9917.1 [15].

To explore the relationship between setting time and other parameters influencing cement hydration, researchers have investigated alternative methods to replace indentation tests. Ha et al. [16] compared indentation tests with rheological measurements to assess the influence of particle size and the inclusion of bismuth oxide on the setting time of MTA and Portland cement. Tsubota et al. [17] examined the setting behavior of luting cements using an ultrasonic method. In this study, we propose the utilization of isothermal calorimetry to evaluate the setting time while simultaneously investigating the hydration kinetics of the cement. By utilizing isothermal calorimetry, we aim to gain more comprehensive insights into the setting behavior of dental cements, addressing the limitations of conventional indentation tests.

Isothermal induction calorimetry is a highly accurate technique widely employed to comprehensively assess the cement hydration process by monitoring the heat flow evolution. Through real-time heat flow measurements, this method allows for precise determination of the exact moment when hydration products are formed, estimation of the material’s workable time, and detailed analysis of the impact of various influencing factors. In this particular case, special emphasis was placed on examining the influence of particle size on the hydration process evolution. Figure 1 exemplifies the rate of hydration by flow heat measurements over time of hydration highlighting the five main typical stages of hydration reaction, that is, I—the initial reaction (dissolution of ions), II—the induction period (dormant state), III—the accelerating period, which is driven by the rapid formation and growth of calcium silicate hydrate (CSH) and portlandite (CH), IV—the decelerating period, and V—the period of slow continued reaction, where the diffusion of ions is progressively reduced, as well as the formation of the hydration products [18]. It is also worth differentiating the initial and final setting times of cement. From a microscopic point of view, the initial setting is the time when the cement’s hydration products begin to form a reticulated network of hydrated products, making the cement rigid enough to offer a primary resistance from flowing. The final setting is given when this network of hydration products is strong enough to support loads.

According to Weiss [19], the initial setting is located near the lowest point of the accelerating period (stage III), while the final setting time is located near the uppermost point (transition from stage III to IV). Sandberg and Liberman [20] defined the setting times according to relative values of temperature rise of the samples, where the initial set corresponds to 21% (near to the lowest point) and the final set to 42% (near to the midpoint). Hu and Ge [21] also used the lowest point (initial set) and midpoint (final set) to study the setting times of mortars influenced by cement (Portland) fineness and water–cement ratio. Tayler [22], Mehta and Monteiro [18], and Neville [23] reported final setting times located at the uppermost point while Ramachandran [24], Jawed et al. [25], and Viecili et al. [26] reported them at the midpoint. According to ASTM C1679 standard [16], the initial set occurs halfway through acceleration period (midpoint) while the final setting occurs when the heat flow or temperature is maximum (uppermost point).

In this particular context, the primary objective of this study was to conduct a comparative analysis between the results obtained from Gillmore indentation tests (particularly the final setting time) and calorimetry tests for biocements with different particle sizes. Previous studies have consistently emphasized the significant impact that particle size exerts on multiple facets of bioceramic cements, encompassing hydration kinetics, microstructure, and physical properties [20,27,28,29,30]. However, notably, the correlation between these two techniques has not been explored or investigated thus far, as per the authors’ knowledge.

Furthermore, this study provides insights into the use of the rarely explored technique of isothermal induction calorimetry in endodontic cements. The utilization of this technique could provide valuable information about the hydration kinetics, heat evolution, and thermal behavior during the curing process of endodontic cements. Implementing it can enhance the comprehension of cement setting, with the goal of optimizing mix design and enhancing properties in both fresh and hardened states.

## 2. Materials and Methods

The as-received C_3_S powder (supplier: Mineral Research Processing, Meyzieu, France) was subjected to dry milling using a ZrO_2_ mill Retsch PM 100 (Retsch, Haan, Germany) with ZrO_2_ microspheres with diameters of 0.3 mm and 0.4 mm. Two different milling times, 1 h and 2 h, were investigated in this study. The selection of these specific milling durations was based on a previously published work [31]. The obtained particle size range in this study was found to be compatible with commercial root sealer cements.

The particle sizes of the as-received and milled samples were determined using laser diffraction with a particle size analyzer instrument Cilas 1064 (Cilas, Orleans, France). The surface areas were measured using the BET method (Brunauer–Emmet–Teller) with a Quantachrome NOVA 1200e instrument (Quantachrome Instruments, Boynton Beach, FL, USA). Additionally, scanning electron microscope (SEM) analysis was conducted on the milled sample powders to investigate their characteristics. X-ray diffraction (XRD) analysis was performed using a Shimadzu XRD-6000 instrument (Shimadzu Corporation, Kyoto, Japan) to confirm the presence of Ca_3_SiO_5_ (purity).

The indentation setting time tests were conducted in accordance with ISO 6876:2012 (International Organization for Standardization, 2012) [1]. Six samples of each C_3_S powder (as received, milled for 1 h, and milled for 2 h) with a water/solid ratio of 0.66 were mixed. The number of samples was carefully selected to ensure result accuracy. Consideration was given to the possibility of outliers and the limited quantity of available material. Thus, the chosen sample size allowed for obtaining a representative and reliable average while also enabling the proper execution of all other tests.

The mixing procedure aimed to achieve a homogeneous mixture, free of any lumps or inhomogeneities. Samples were placed in plaster molds with disc-shaped cavities measuring 10 mm in internal diameter and 1 mm in height. Before pouring the cementitious mix into the molds, they were meticulously cleaned and dried to prevent any contamination or adhesion issues. Once the molds were prepared, the cementitious mix was poured into each mold until it reached the top edge. To create a flat and even surface, a straight-edge tool was carefully run over the top of the mold. This step ensured that the surface of the sample was smooth and free of irregularities, which is crucial for accurate indentation testing. Once the surface was leveled and excess material removed, the samples underwent indentation testing. This involved using a 100 ± 0.5 g Gillmore needle with a cylindrical tip, measuring 2.0 ± 0.1 mm in diameter. The needle was applied vertically to the horizontal surface of the cement. Each indentation was performed at 15-min intervals. This process was repeated until the indentation mark was no longer visible, indicating the final setting time. Three samples of each cement were tested at room temperature (23 °C).

Isothermal induction calorimetry analysis was performed using a TAM Air Isothermal Calorimeter (TA Instruments, New Castle, DE, USA). Samples with the same water/solid ratio of 0.66 were manually mixed for 2 min, placed in glass ampoules (approximately 5 g), and monitored using the equipment for 72 h at 23 °C. Measurements were automatically recorded by a data logger connected to a computer.

## 3. Results and Discussion

### 3.1. Features of Anhydrous Powders

Table 1 presents the average particle sizes and surface areas of the as-received sample, as well as the samples milled for 1 h and 2 h. As anticipated, the increase in milling time resulted in a considerable reduction in the D_50_ percentile and increase in the surface area.

Figure 2 displays the X-ray diffraction (XRD) patterns of the samples. The majority of the peaks correspond to the Ca_3_SiO_5_ phase, except for the peak at 32.5°, which corresponds to Ca_2_SiO_4_. The presence of small amounts of C_2_S is common when there is a slight deviation from the stoichiometry of C_3_S (73.7 wt.% CaO/26.3 wt.% SiO_2_). However, there were no indications of free-CaO, which typically forms during the cooling/quenching process of C_3_S.

Figure 3 depicts SEM micrographs showcasing the powder samples. While some agglomerated particles are observable in the images, the measured particle sizes align reasonably well with those acquired through the particle size analyzer. Regarding morphology, it is evident that all particles display irregular shapes.

### 3.2. Setting Behavior of Hydrated Samples

Figure 4 presents the relationship between setting times obtained from ISO 6876 indentation testing and isothermal induction calorimetry. Figure 4a demonstrates a strong correlation between the final setting time determined by indentation and the midpoint of the acceleration period on the calorimetry curve, with time deviations of less than 12% for all particle sizes.

Table 2 summarizes the setting time data obtained from both methods. By comprehending the relationship between milling time, particle size, and setting behavior, clinicians and researchers can customize biocement formulations to match specific clinical needs. Striking a balance in particle size enables optimal handling and working times during procedures, ensuring robust bonding and compatibility with tissues. As noted, the initial hour of milling proves to be more effective in reducing the setting time, as evidenced by both indentation and calorimetry analyses. As the milling time increases, the effectiveness in reducing the setting time diminishes. Therefore, controlling the particle size becomes a crucial factor in optimizing the setting behavior of biocements. Smaller cement particles have a larger total surface area compared to larger particles for the same mass of cement. As a result, when water is added, more water–cement interactions can occur on the surfaces of the smaller particles. This leads to a higher rate of hydration reactions, allowing the cement to set and gain strength more rapidly. The increased surface area of smaller particles also facilitates the nucleation and growth of CSH and CH crystals, promoting the formation of a denser and more interconnected microstructure. This dense microstructure contributes to the early strength development of the cement, as it offers more efficient load transfer and enhanced mechanical properties. This phenomenon can be followed by the heat generated over time, discussed further.

Figure 4b displays the cumulative heat generated during the indentation setting time, from the beginning until the midpoint and uppermost point of the acceleration curve. Due to the proximity of the indentation setting times and the midpoint of the calorimetry curve, the cumulative heat flows are also closely related. In Figure 4c, a notable linear correlation is evident between the setting times determined by both methods. This observation implies that the results from the indentation test can serve as a reliable means to estimate both the midpoint and the uppermost point of the heat flow curve. This correlation enables a comprehensive understanding of paste behavior during the acceleration period. Similarly, Figure 4d demonstrates the relationship between the heat flow of the samples and the final indentation setting times. There is an excellent linear correlation between the midpoint of the calorimetry curve and the indentation results. However, the heat flow shows an exponential behavior when considering the entire interval until the end of the acceleration period (uppermost point), particularly evident for the 2 h-milled C_3_S with a shorter indentation setting time of 4.2 h.

Figure 4e displays the cumulative heat flow, which follows a similar pattern with a linear increase in heat for the midpoints and an exponential increase for shorter final setting times. By utilizing the correlations depicted in Figure 4c (time), Figure 4d (heat flow), and Figure 4e (cumulative heat), it is possible to outline the acceleration period through calorimetry from indentation tests results. This assumption holds true for powders with different particle sizes, but only when the same water/solid ratio (0.66) is maintained.

Figure 4f illustrates the cumulative heat generated by the three samples over an extended period of 3 days. At the end of the deceleration period of the 2 h-milled sample (11 h of hydration), the cumulative heat reaches 356 J/g, which is approximately 58% higher than that of the 1 h-milled sample (225 J/g) and 159% higher than the as-received sample (137 J/g). It is noteworthy that the 1 h-milled sample required 51 h of hydration to achieve the same cumulative heat as the 2 h-milled sample with 11 h of hydration, while the as-received sample did not reach this level within the 3-day analysis period.

Figure 4g allows for a comparison of the cumulative heat of the samples at different hydration periods. For instance, the cumulative heat of the 2 h-milled sample at 24 h is similar to that of the 1 h-milled sample at 72 h, and the cumulative heat of the 1 h-milled sample at 24 h is comparable to that of the as-received sample at 72 h.

The C_3_S hydration process triggers spontaneous exothermic reactions, primarily associated with the formation of CSH (calcium silicate hydrate) and CH (calcium hydroxide). Consequently, a higher heat flow during a specific period corresponds to a greater amount of CSH and CH formed. While the Ca/Si ratio in CSH may exhibit variations during the hydration process, especially in the early stages, it can be estimated using Equation (1). The enthalpy of reaction (ΔH_R_) is calculated as the difference between the total enthalpies of formation (ΔH_F_) of the products and reactants. For C_3_S, ΔH_R_ is determined to be −558 J/g [32].


(1)
$$\underset{228.3\;g}{\underset{︸}{C_{3}S}}+\underset{70.3\;g}{\underset{︸}{3.9H}}\to \underset{202.3\;g}{\underset{︸}{C_{1.7}SH_{2.6}}}+\underset{96.3\;g}{\underset{︸}{1.3CH}}\;\;\;\;\;\;\;\;\;\;\;\;\;\;\;\;\Delta H_{R}=-558\;J/g$$


In line with previous studies [32,33,34,35], it can be assumed that the dissolution and precipitation process associated with the hydration of C_3_S occur simultaneously. Therefore, by considering the heat released from the complete reaction of C_3_S (as given in Equation (1)) and the cumulative heat measured by calorimetry—which represents the partial reaction of C_3_S (as shown in Figure 4b,f)—it becomes feasible to estimate the degree of hydration and the mineralogy of C_3_S during the hydration process, proportional to the weights of the products and reactants.

Figure 5 presents the estimated phase and pore solution contents for the samples hydrated for up to 24 h, during which the majority of the hydration reactions take place. Table 3 summarizes the degree of hydration and the proportion of continuous phases for the respective indentation setting times (5.5 h, 4.5 h, and 4.2 h for the as-received, 1 h-milled, and 2 h-milled samples, respectively) as well as after 11 h and 24 h of hydration.

At this moment, it is worth providing a brief explanation concerning the chemical reactions between C_3_S and water, which can be followed by the graphical abstract, aiding in the interpretation of data from Figure 5 and Table 3.

The beginning of the cement hydration process can be divided into two main stages: dissolution and precipitation. Within seconds to minutes after the commencement of cement hydration, the anhydrous C_3_S grains release Ca into the aqueous mixture, known as the pore solution. Within minutes, the nuclei of CSH can form and gradually grow. During this phase, the paste remains in a dormant state, entering what is called the induction period. After 1–2 h (depending on particle size and temperature), the concentration of Ca ions increases until the pore solution becomes supersaturated. This state energetically favors some Ca ions to precipitate into solid phases, mainly CSH and CH, rather than remaining dissolved. The precipitation of hydration products reduces the supersaturation of the pore solution and allows the C_3_S grains to continue dissolving further. Consequently, the hydration products grow at an accelerated rate, eventually forming a thick enough layer around the C_3_S particles, leading to interactions with other particles. These interactions form a cross-linked network of hydration products that are responsible for preventing indenter penetration, coinciding with the midpoint of the acceleration period, as previously observed [22,36,37].

In this context, it is noteworthy that the reactivity and the speed of cross-linked network formation are higher when the particle sizes of C_3_S are smaller. As the layer around the C_3_S grains thickens, the diffusion of Ca from the pore solution reduces, along with the formation and growth of CSH. This phenomenon marks the end of the acceleration period and the beginning of the deceleration period. The C_3_S hydration is thus a continuous process whereby unhydrated C_3_S is transformed into solid hydration products, with the pore solution acting as a necessary intermediate phase between the two solid states [22,36,37].

The results indicate a clear correlation between the setting time and the hydration degree of the samples. The end of the setting time corresponds to a hydration degree of approximately 10 ± 2%, where around 10–15 wt.% of the C_3_S has reacted, resulting in the formation of approximately 7–10 wt.% of CSH. This implies that the final indentation setting time occurs when 10–15 wt.% of the hydrated products have formed, creating a network structure that prevents needle penetration.

After approximately 11 h, which aligns with the end of the deceleration period for the 2 h-milled sample (as depicted in Figure 5c), a plateau is observed as the hydration degree surpasses 63%. Beyond this point, further growth becomes slower. This indicates that around 63% of the C_3_S has reacted with water to form CSH and CH. Subsequently, there are no significant changes, implying a considerable reduction in the rate of hydration product formation. This is attributed to the thick layer of hydrated products formed on the surface of the anhydrous C_3_S, which hampers the diffusion of calcium from the pore solution and impedes the formation and growth of CSH.

For the 1 h-milled sample (Figure 5c), the hydration degree plateau is reached at approximately 24 h, corresponding to a hydration degree of 55%. These findings suggest that the end of the deceleration period occurs when the sample reaches a hydration degree close to 60 ± 5%, with CSH constituting approximately 45 ± 3 wt.% of the solid phase.

## 4. Conclusions

This study aimed to investigate the setting behavior of C_3_S, the main component of bioceramic cements, by varying particle sizes and water–solid ratios. We compared two methods for determining the final setting time of bioceramic cements: the commonly used indentation test, a simple and cost-effective approach, and calorimetry, a technique providing detailed information on the hydration process of hydraulic cements but requiring specialized equipment and training to operate.

It was observed that the final setting times obtained through the indentation method closely approximate the midpoints of the acceleration curves from calorimetric analysis, with time deviations of less than 12% for powders with different particle sizes.

The results also revealed a significant influence of particle size on the hydration rate. As the particle size decreased from 5.40 μm (as-received) to 0.82 μm (2 h-milled), the setting time reduced from 5.5 h to 4.2 h (−24%), and the end of the deceleration period decreased from 15.3 h to 11 h (−28%). The reduction in setting times and the end of the acceleration period displayed a relatively linear relationship with the decrease in particle size. It is important to highlight that the tests were conducted with particle size variations close to those found in commercially available root sealer cements. For predominantly nanoscale materials, for instance, the results may exhibit different behavior, and therefore, this correlation needs to be further validated in future studies.

The reduction in particle size caused a substantial increase in reactivity, especially in the first few hours. While the as-received sample reacted approximately 32% at the end of the deceleration period at 15.3 h, the 2-h milled sample reacted 63% and with the deceleration period ending at 11 h of hydration (the reactivity is, therefore, doubled with 2 h of milling). Consequently, it can be inferred that biocements with smaller C_3_S particles present much greater mechanical resistance sooner.

By analyzing the curve fits obtained from calorimetric analysis and indentation testing, it becomes possible to estimate the heat flow and cumulative heat of the samples solely based on the indentation tests. Additionally, this methodology allows for the characterization of the cement’s mineralogy at the time when the final setting time of biocement is achieved, as well as at the end of the acceleration period (uppermost point in the calorimetric curve).

From a microscopic perspective, the final set can be attributed to the formation of a primary cross-linked network of hydration products capable of supporting the indentation needle. This network typically forms when approximately 10–15 wt.% of the hydrated products have been formed.

Future research can associate calorimetric analysis with the mineralogy of single-phase biocements and its other intriguing properties, such as mechanical or biological characteristics. However, for comprehensive analyses of multiphase materials, the preferred method would continue to be the utilization of X-ray diffraction in conjunction with the Rietveld method.

## Figures and Tables

**Figure 1 bioengineering-11-00036-f001:**
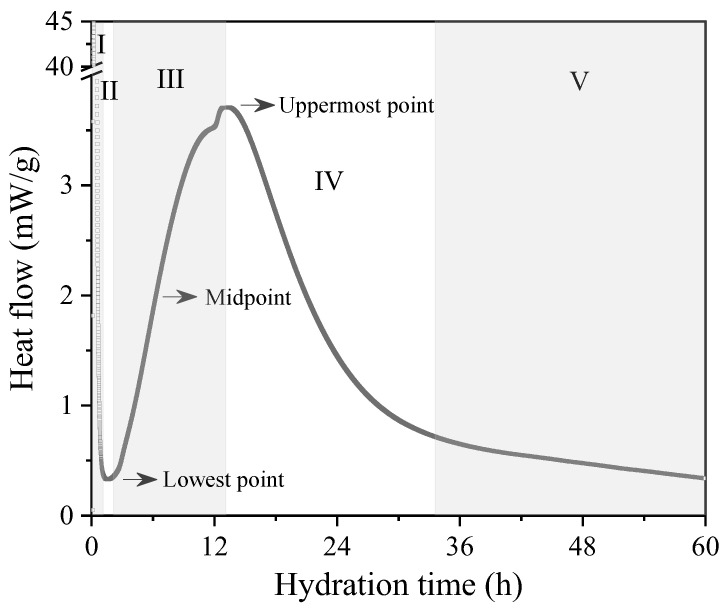
Typical heat flow curve with the five main hydration stages. I—initial reaction; II—induction period; III—acceleration period; IV—deceleration period; and V—period of slow reaction. The lowest point, midpoint, and highest point of the acceleration period are also indicated as references for determining the initial and final setting times.

**Figure 2 bioengineering-11-00036-f002:**
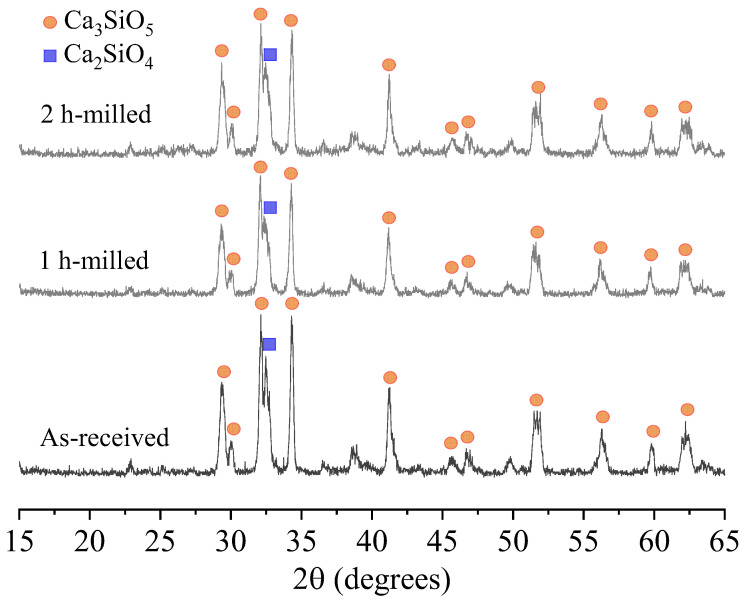
XRD patterns of the samples. Ca_3_SiO_5_ JCPDS card 01-070-8632; Ca_2_SiO_4_ JCPDS card 01-077-0388.

**Figure 3 bioengineering-11-00036-f003:**
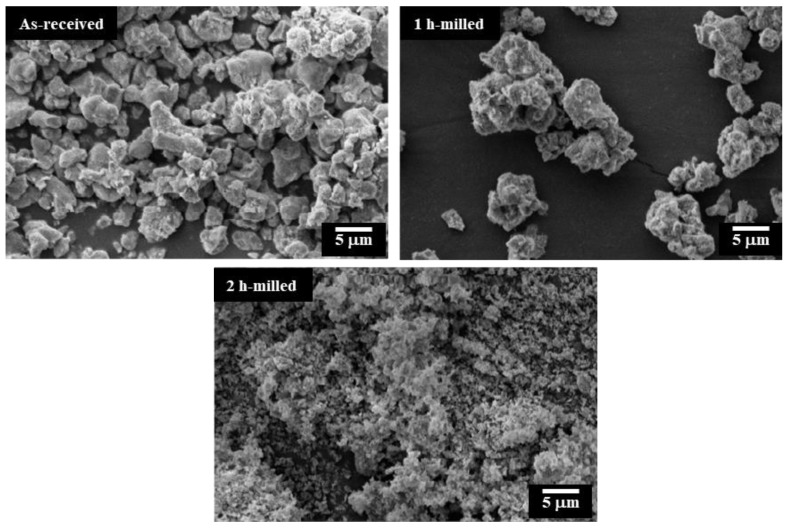
SEM images of the sample powders.

**Figure 4 bioengineering-11-00036-f004:**
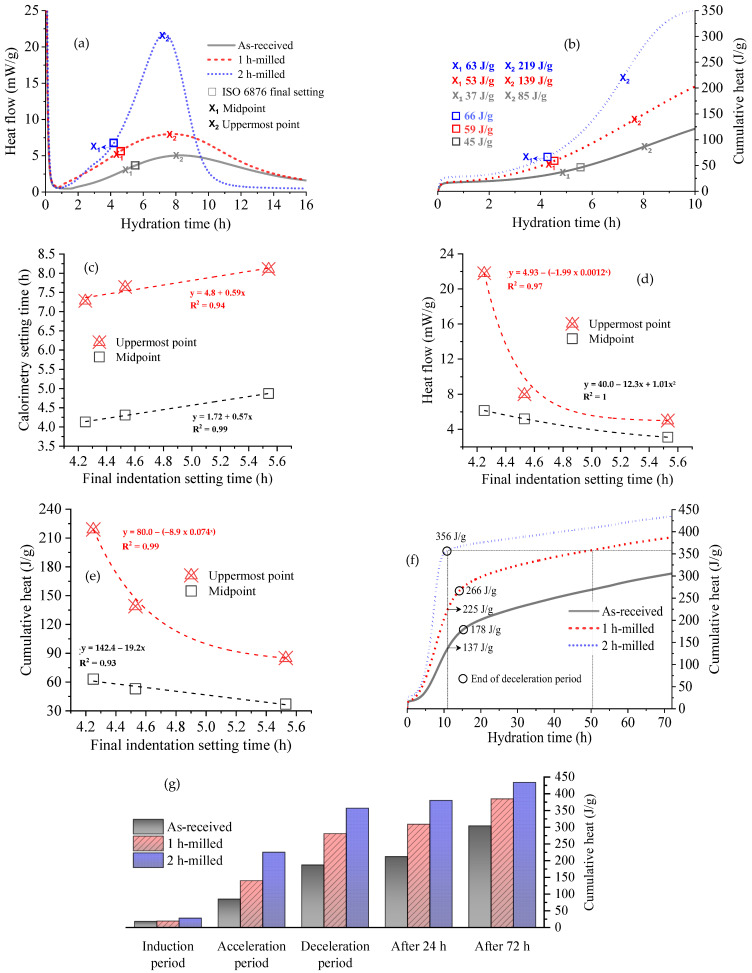
(**a**) Heat flow curve, (**b**) cumulative heat in first 10 h, (**c**) correlation between calorimetry and final indentation setting times, (**d**) correlation between heat flow and final indentation setting time, (**e**) correlation between cumulative heat and final setting time, (**f**) cumulative heat for 72 h of hydration, and (**g**) cumulative heat at the ending of different hydration periods.

**Figure 5 bioengineering-11-00036-f005:**
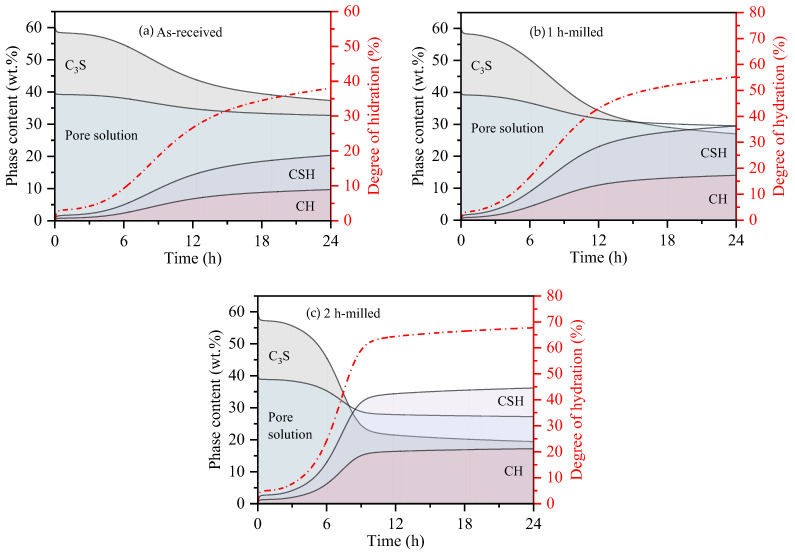
Estimation of phase content and degree of hydration of samples hydrated in the first 24 h for (**a**) as-received, (**b**) 1 h-milled, and (**c**) 2 h-milled.

**Table 1 bioengineering-11-00036-t001:** Average particle size and surface area of the samples.

Sample Preparation	Average Particle Size (D_50_)	Surface Area (m^2^/g)
As received	5.40 µm	4.84 m^2^/g
1 h-milled	2.79 µm	12.22 m^2^/g
2 h-milled	0.82 µm	15.84 m^2^/g

**Table 2 bioengineering-11-00036-t002:** Data summary of setting times by indentation test and isothermal calorimetry. The numbers in parentheses indicate the variation of the outcome concerning the previous sample.

Sample	Indentation Final Setting Time (h)	Midpoint of the Acceleration Period (h)	Uppermost Point of the Acceleration Period (h)
As received (5.40 µm)	5.5	4.87	8.11
1 h-milled (2.79 µm)	4.5 (18.2%)	4.31 (11.5%)	7.64 (5.8%)
2 h-milled (0.82 µm)	4.2 (6.7%)	4.13 (4.2%)	7.28 (4.7%)

**Table 3 bioengineering-11-00036-t003:** Estimation of degree of hydration and solid phase’s proportion from calorimetry data.

Sample	* Final Indentation Time/ ** Time of Hydration (h)	Degree of Hydration (%)	Solid Phase’s Proportion (wt.%)		
			CSH	CH	Unhydrate C_3_S
As-received	5.5 *	8	7	3	90
1 h-milled	4.5 *	10	9	4	87
2 h-milled	4.2 *	11	10	5	85
As-received	11 **	24	20	10	70
1 h-milled		40	31	15	54
2 h-milled		63	47	22	31
As-received	24 **	38	30	14	56
1 h-milled		55	42	20	38
2 h-milled		68	50	23	27

## Data Availability

Data will be made available upon request.

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
