# Peer review of "Impact of Particle Size on the Setting Behavior of Tricalcium Silicate: A Comparative Study Using ISO 6876 Indentation Testing and Isothermal Induction Calorimetry"

_bioengineering, 2023, doi:10.3390/bioengineering11010036_

Round 1

Reviewer 1 Report

Dear Authors

This study examines the impact of particle size on the setting behaviour of tricalcium silicate powders. Although the topic and results are very interesting, important details are needed in the material and methods section in order to improve the ms. In addition, the authors did not follow the manuscript sections required by the journal. For these and other reasons I suggest a major revision before publication. Comments are within the PDF file.

The english sounds good. However, I suggest to simplify some sentences, mainly the longest one. 

Author Response

Dear Reviewer,

Thank you for your careful reading and contributions, and for allowing us to clarify important issues raised. The comments and suggestions were very constructive for the improvement of our manuscript. We feel that we have addressed the reviewer's concerns and, as a result, the quality of the manuscript has improved. 

We have carefully addressed all the reviewer's comments and have made the necessary revisions accordingly. The viscosity of the material has been cited, highlighting its potential influence on the penetration capability of cements. Additionally, we have provided a more detailed explanation for our decision to use a flow heat diagram, emphasizing the advantages of utilizing heat flow measurements in our study.

Regarding the sample size, we have included a clear rationale for choosing six samples to justify this selection. Furthermore, we have now explicitly specified the process for adjusting the superficial layer of the samples.

The indentation setting time tests were conducted in accordance with ISO 6876:2012 (International Organization for Standardization, 2012) [1]. Six samples of each C3S pow-der (as-received, milled for 1 h, and milled for 2 h) with a water/solid ratio of 0.66 were mixed. The mixing procedure aimed to achieve a homogeneous mixture, free of any lumps or inhomogeneities. Samples placed in plaster molds with disc-shaped cavities measuring 10 mm in internal diameter and 1 mm in height. Before pouring the cementitious mix into the molds, they were meticulously cleaned and dried to prevent any contamination or adhesion issues. Once the molds were prepared, the cementitious mix was poured into each mold until it reached the top edge. To create a flat and even surface, a straight-edge tool was carefully run over the top of the mold. This step ensured that the surface of the sample was smooth and free of irregularities, which is crucial for accurate indentation testing. After surface leveling and excess material removal, the samples were allowed to rest and set at room temperature (23°C) for a specified period before conducting the indentation test. This ensured that the cementitious material achieved the appropriate level of initial setting. The samples were subjected to indentation using a 100 ± 0.5 g Gillmore needle with a cylindrical tip measuring 2.0 ± 0.1 mm in diameter, applied vertically to the horizontal surface of the cement. Each indentation was performed at 15-min intervals. This process was repeated until the indentation mark was no longer visible, indicating the final setting time. Three samples of each cement were tested at room temperature (23°C).

In response to the suggestion for better analysis, we agree that all plots in Figure 4 should be inserted in the same page, making it easier to interpret the results. We appreciate your understanding that the art department will be responsible for integrating the plots, which will undoubtedly enhance the visual representation of the data.

Lastly, we have thoroughly followed the instructions for Authors provided by the journal, ensuring that our manuscript adheres to the "Research Manuscript Sections" guidelines.

Once again, we thank the reviewer for their constructive feedback, and we believe that these improvements have strengthened the quality of our manuscript. We are now ready to submit the revised version for further consideration.

Reviewer 2 Report

The paper contains the experiment study of the impact of particle size on the setting behavior of Tricalcium Silicate. However, there are still some questions that the authors may need to answer:

1. Introduction. “their ability to penetrate dentinal tubules and lateral canals can be influenced by factors like cement mineralogy, particle size…”. Since there are many factors affecting the biocompatibility of these cements, which factor has the greatest impact. Whether the impact of particle size is large enough, so the innovation of this paper needs to be highlighted.

2. According to the published literature standards, there are different definitions of the initial setting time and the final setting time. Which one is used in this paper, please explain.

3. Materials and methods. What is the basis of one hour and two hour milling of powders, please explain.

4. In Figure 2. The deposited compounds were identified only in the samples with 2 h-milled. What compounds are founded in the specimens with 1 h-milled compared to the unaged sample, is it the same as the specimens with 2 h-milled?

5. “the percentage reduction in setting times and the end of the acceleration period showed a reasonably linear relationship with particle size reduction…”. The linear relationship between setting times and particle size needs more experiments to prove. In this paper there are only three particle sizes, and it is reasonable to set an interval for the particle sizes when conducting the discussion.

6. The grammar and writing of the manuscript need to be further improved.

The grammar and writing of the manuscript need to be further improved.

Author Response

Dear Reviewer,

Thank you for your careful reading and contributions, and for allowing us to clarify important issues raised. The comments and suggestions were very constructive for the improvement of our manuscript. We feel that we have addressed the reviewer's concerns and, as a result, the quality of the manuscript has improved. Changes are highlighted throughout the manuscript (yellow for added text and blue for improved language). This document follows our responses to reviewers' comments and questions.

The paper contains the experiment study of the impact of particle size on the setting behavior of Tricalcium Silicate. However, there are still some questions that the authors may need to answer:

  1. Introduction. “their ability to penetrate dentinal tubules and lateral canals can be influenced by factors like cement mineralogy, particle size…”. Since there are many factors affecting the biocompatibility of these cements, which factor has the greatest impact. Whether the impact of particle size is large enough, so the innovation of this paper needs to be highlighted.

Response: We acknowledge that all these several factors may substantially influence the biocompatibility. In this study, however, we specifically focused on examining the impact of particle size on setting time through different techniques, and indirectly on the mineralogical phases through heat flow measurements. With this approach, we aim to provide a clear and targeted investigation into the interplay between these variables.

To address the concerns raised by the reviewers and enhance the clarity of our study, we have rephrased the sentence to focus explicitly on the influence of particle size on setting time. Additionally, we have emphasized the use of heat flow measurements to indirectly examine its impact on the mineralogical phases. By adopting this approach, we aim to provide a more targeted investigation, shedding light on the specific relationship between particle size, setting time, and mineralogical changes, which are crucial aspects in assessing biocompatibility.

“As biocompatibility is a critical factor influenced by cement mineralogy, the clinical performance, especially in biocement applications, hinges on particle size and temperature. These factors play a crucial role in determining the cement's viscosity and setting time, directly impacting its ability to effectively penetrate and fill dentinal tubules and lateral canals (1,4–6). Root canal sealers can have different viscosities, commonly determined ac-cording to flow measurements according to ISO 6876/2012 (7). Certain endodontic sealers are specifically designed with higher fluidity to facilitate enhanced penetration into intricate canal systems, while others may possess lower fluidity to optimize handling characteristics. The ISO 6876/2012 recommend a minimum of 17 mm. In a study conducted by Torres et al. (8), different root canal sealers were found to exhibit values near 20 mm.

While the influence of temperature is challenging to precisely predict due to variations in environmental and individual body temperatures, controlling particle size represents another means of manipulating the setting time to meet ideal clinical standards. The particle size of root canal sealers can vary based on the specific type used. Root sealer cements typically contain particles size under 10 μm (9,10), which aids in better flow and penetration into complex canal systems and contributes to achieving a reasonable setting time. From a practical standpoint, an excessively short setting time can negatively affect the workability of the cement, while an excessively long setting time can compromise cement solubility and disintegration in tissue fluid, thereby impairing sealing ability and bioactivity. In clinical practice, the setting time of endodontic cements is typically less than 30 minutes, normally achieved with chemical admixtures accelerating the setting time.”

“[…] this study provides insights into the use of the rarely explored technique of isothermal induction calorimetry in endodontic cements. The utilization of this technique could provide valuable information about the hydration kinetics, heat evolution, and thermal behavior during the curing process of endodontic cements. Its application could contribute to a better understanding of the cement setting, enabling formulation optimization and improvement of their physical properties.”

  1. According to the published literature standards, there are different definitions of the initial setting time and the final setting time. Which one is used in this paper, please explain.

Response: The primary focus of this work was indeed on the final setting time. While the initial setting time is associated with the initial loss of plasticity and the formation of the first hydration products, the final setting time pertains to the complete hardening of the material, characterized by the formation of a cross-linked network of hydration products. This process can be followed by heat flow measurements, which provide valuable insights into the setting kinetics and the development of the material's strength.

We have rewritten sentences and brought additional information to enhance the clarity and understanding of the text.

“Setting time can be divided into two key phases: initial and final setting times. The initial setting time is the duration from the moment water is added to the cementitious mixture until the material starts to lose its plasticity and begins to harden. This stage is particularly important as it determines the window during which the material can be placed, molded, or worked on before it becomes too rigid. On the other hand, the final setting time represents the period from the introduction of water to the cementitious mixture until the material achieves its full hardening and load-bearing strength. Knowing the final setting time is essential for determining when the hardened material can be subjected to stresses safely. This occurs through the formation of a cross-linked network of hydration products, which can be evaluated and compared using indentation testing and calorimetric analysis.”

[…]

“In this particular context, the primary objective of this study was to conduct a comparative analysis between the results obtained from Gilmore indentation tests (particularly the final setting time) and calorimetry tests for biocements with different particle sizes.”

  1. Materials and methods. What is the basis of one hour and two hours milling of powders, please explain.

Response: We appreciate your inquiry regarding the milling times used in our study. The selection of one hour and two hours of milling duration was based on careful considerations to ensure optimal results.

Firstly, shorter milling times may not be sufficient to adequately reduce the particle size. Milling requires adequate time for the particles to be effectively broken down to the desired size. If the milling duration is too short, it may result in larger particles, compromising the desired characteristics and properties of the final material.

Conversely, longer milling times can lead to excessively reduced particle sizes. Extremely fine particles might increase the need for additional water during processing, potentially affecting other process steps or the final application of the material.

It is important to note that the choice of milling times can be influenced by the specific nature of the material being processed. Different materials may require different milling durations to achieve the ideal particle size.

We also referenced a previously published work that successfully utilized similar milling times with the same type of mill. The obtained particle size range in this study was found to be compatible with commercial root sealer cements.

To address your question in the manuscript, we have incorporated the new text in the Materials and Methods section. If you have any further questions or points you would like to discuss, please feel free to reach out to us. We are available to provide further clarification as needed.

“The as-received C3S powder (supplier: Mineral Research Processing, Meyzieu, France) was subjected to dry milling using a ZrO2 mill (Retsch PM 100, Germany) with ZrO2 microspheres having diameters of 0.3 mm and 0.4 mm. Two different milling times, 1 h and 2 h, were investigated in this study. The selection of these specific milling durations was based on a previously published work (31). The obtained particle size range in this study was found to be compatible with commercial root sealer cements.”

  1. Modolon HB, Inocente J, Bernardin AM, Klegues Montedo OR, Arcaro S. Nanostructured biological hydroxyapatite from Tilapia bone: A pathway to control crystallite size and crystallinity. Ceram Int [Internet]. 2021 Oct;47(19):27685–93. Available from: https://linkinghub.elsevier.com/retrieve/pii/S0272884221019386

  1. In Figure 2. The deposited compounds were identified only in the samples with 2 h-milled. What compounds are founded in the specimens with 1 h-milled compared to the unaged sample, is it the same as the specimens with 2 h-milled?

Response: Indeed, the compounds are identical. We have distinguished and identified the phases present in the other specimens in Figure 2 to enhance clarity.

  1. “the percentage reduction in setting times and the end of the acceleration period showed a reasonably linear relationship with particle size reduction…”. The linear relationship between setting times and particle size needs more experiments to prove. In this paper there are only three particle sizes, and it is reasonable to set an interval for the particle sizes when conducting the discussion.

Response: Thank you for the observation. In response to the reviewer's feedback, we have added a note regarding the correlation between setting time, calorimetry, and particle size in the "Conclusion" section to address the raised concern.

“The reduction in setting times and the end of the acceleration period displayed a relatively linear relationship with the decrease in particle size. It is important to highlight that the tests were conducted with particle size variations close to those found in commercially available root sealers cements. For predominantly nanoscale materials, for instance, the results may exhibit different behavior, and therefore, this correlation needs to be further validated in future studies.”

  1. The grammar and writing of the manuscript need to be further improved.

Response: Ok. We have made additional enhancements (in blue) to improve the readability of the manuscript by using artificial intelligence.

Round 2

Reviewer 1 Report

Dear Authors

Although your improvements have strengthened the quality of the manuscript, just few information are required before the publication. Firstly, although you answered "Regarding the sample size, we have included a clear rationale for choosing six samples to justify this selection" I do not find this information in the text. Moreover, the Discussion section misses, is there any reason?

Author Response

Thank you for your valuable feedback. We apologize for the oversight in not explicitly providing the rationale for choosing six samples in the text. To address this, we have revised the manuscript to include a clear explanation in the Materials and Methods section, outlining the basis for selecting six samples to justify our choice adequately.

“The number of samples was carefully selected to ensure result accuracy. Consideration was given to the possibility of outliers and the limited quantity of available material. Thus, the chosen sample size allowed for obtaining a representative and reliable average while also enabling the proper execution of all other tests.”

Regarding the missing Discussion section, we apologize for any confusion. We have indeed included the discussion of the findings, but it appears there might have been an oversight in not explicitly mentioning it in the manuscript. To address this, we have now rectified the situation by changing the section 4 heading from "Results" to "Results and Discussion."

Reviewer 2 Report

Here is my feedback on the revised manuscript which I had previously recommended for major revision.

I can see that the authors have thoroughly responded to the review questions and made the necessary changes to improve the quality of the manuscript. However, there are still a few questions that need to answer:

1. Reference references should use hyperlinks to facilitate rapid positioning.

2. On page 7, The size of Figure 4 needs to be adjusted to make it on the same page.

3. On page 6, The Setting behavior of hydrated samples should be discussed in more detail.

Please also double-check all other references to ensure that similar issues have not been overlooked.

The grammar and writing of the manuscript need to be further improved.

Author Response

Here is my feedback on the revised manuscript which I had previously recommended for major revision.

I can see that the authors have thoroughly responded to the review questions and made the necessary changes to improve the quality of the manuscript. However, there are still a few questions that need to answer:

  1. Reference references should use hyperlinks to facilitate rapid positioning.

Response: We appreciate your feedback and are glad to hear that the revisions have improved the manuscript's quality.

Regarding the reference formatting, we have now provided hyperlinks to facilitate easier navigation and quicker access to the referenced sources.

  1. On page 7, The size of Figure 4 needs to be adjusted to make it on the same page.

Response: To address your concern about the setting behavior of hydrated samples, we have expanded the discussion in more detail, delving into the mechanisms related to the setting behavior and associated phenomena. This provides a comprehensive understanding of the experimental results and their implications.

  1. On page 6, The Setting behavior of hydrated samples should be discussed in more detail.

Response: To address your concern about the setting behavior of hydrated samples, we have expanded the discussion in more detail, delving into the mechanisms related to the setting behavior and associated phenomena. This provides a comprehensive understanding of the experimental results and their implications. 

Please also double-check all other references to ensure that similar issues have not been overlooked.

Comments on the Quality of English Language

The grammar and writing of the manuscript need to be further improved

Response:

As for the quality of English language, we have made further improvements to enhance the grammar and overall writing style using artificial intelligence.

We sincerely thank you for your valuable input, which has helped us strengthen the manuscript further. We have made every effort to address all the issues raised and ensure a comprehensive and well-presented research document.